Sex impacts pain behaviour but not emotional reactivity of lambs following ring tail docking

Marini Danila danila.marini@adelaide.edu.au 1 2
Monk Jessica E. 1 3
Campbell Dana L.M. 1
Lee Caroline 1
Belson Sue 1
Small Alison 1
1 Agriculture and Food, The Commonwealth Scientific and Industrial Research Organisation , Armidale , New South Wales , Australia
2 Davies Livestock Research Centre, School of Animal and Veterinary Science, The University of Adelaide , Adelaide , South Australia , Australia
3 School of Environmental and Rural Science, The University of New England , Armidale , New South Wales , Australia
McElligott Alan
Electronic publication date: 2023 Mar 28
Publication date: 2023
Volume: 11
Electronic Location ID: e15092
Received 2022 Nov 14; Accepted 2023 Feb 27
Copyright: ©2023 Marini et al.
Copyright year: 2023
Copyright holder: Marini et al.
License: This is an open access article distributed under the terms of the Creative Commons Attribution License, which permits unrestricted use, distribution, reproduction and adaptation in any medium and for any purpose provided that it is properly attributed. For attribution, the original author(s), title, publication source (PeerJ) and either DOI or URL of the article must be cited.
License URL: https://creativecommons.org/licenses/by/4.0/

Keywords: Animal welfare, Sheep, Acute pain, Attention bias, Tail docking, Affective state, Sex, Startle

Funding: The authors received no funding for this work.

==============================
Studies in humans have shown sex differences in response to painful events, however, little is known in relation to sex differences in sheep. Understanding sex differences would enable improved experimental design and interpretation of studies of painful procedures in sheep. To examine sex differences in response to pain, 80 lambs were tested across five cohorts of 16. The lambs were penned in groups containing two male and two female lambs with their respective mothers. Lambs were randomly allocated from within each block to one of four treatment groups; FRing–Female lamb, ring tail docked without analgesia, MRing–Male lamb, ring tail docked without analgesia, FSham–Female lamb, tail manipulated and MSham–Male lamb, tail manipulated. Following treatment, lambs were returned to their pen and were video recorded for 45 mins for behavioural observations of acute pain and posture. An hour after treatment, lambs then underwent an emotional reactivity test that consisted of three phases: Isolation, Novelty and Startle. Following treatment, Ring lambs displayed more abnormal postures (mean = 2.5 ± 0.5) compared to Sham lambs (mean = 0.05 ± 0.4, P = 0.0001). There was an effect of sex on the display of acute pain-related behaviours in lambs that were tail docked (P < 0.001), with female lambs displaying more acute behaviours (mean count = +2.2). This difference in behaviour between sexes was not observed in Sham lambs. There was no effect of sex on display of postures related to pain (P = 0.99). During the Novelty and Startle phase of the emotional reactivity test, Ring lambs tended to (P = 0.084) or did (P = 0.018) show more fear related behaviours, respectively. However, no effect of sex was observed. The results of this study indicate that a pain state may alter the emotional response of lambs to novel objects and potential fearful situations. It was also demonstrated that female lambs display increased sensitivity to the acute pain caused by tail docking compared to males.

Introduction

Every year millions of lambs undergo a variety of painful husbandry procedures, such as castration and tail docking, for ease of management and to reduce the risk of fly strike (French, Wall & Morgan, 1994). Currently, producers in Australia have access to pain-relief that can be applied at the time of tail-docking that ameliorates the acute pain phase of the procedure (Small, Marini & Colditz, 2021), although pain-relief use is not mandated in Australia. There is growing evidence that the acute pain of these procedures may manifest into chronic pain, especially when pain relief is not given (Johnston et al., 2022; Viñuela Fernández et al., 2007). This has implications for animals kept in the production system long term (such as ewes and breeding rams). Australia’s sheep industry contains a larger proportion of ewes, compared to wethers and rams, as part of self-replacement flocks for meat and or wool production (East & Foreman, 2011). There are extensive studies that look at pain impact and pain-relief research related to the castration of male lambs (Bonelli et al., 2008; Clark et al., 2011; Colditz, Paull & Lee, 2012; Jongman et al., 2000; Kells et al., 2020) as well as work looking at tail docking in male and female lambs (Small et al., 2020). There is little comparison on the effect of these procedures between male and female lambs.

Previous pain research in young lambs (under 12 days of age) observed age differences in the response to pain that was dependant on the sex of the lamb. In the study by Guesgen et al. (2011) there was no observed difference between males and females in the response to thermal nociception between the ages of 1 to 12 days. However, there was a decrease in sensitivity to thermal nociception in male lambs from 1 to 12 days of age (Guesgen et al., 2011). Sensitivity to thermal nociception increased in the female lambs over time, however this observation between age 1 and 12 days was not significant. It is often reported that females have a higher sensitivity to pain, however males have been shown to be more responsive to analgesic treatments (Mogil, 2020). In some studies, differences were only observed in one sex when comparing the effect of analgesia against a before-analgesia or no-analgesia control group (Mogil, 2020).

As a prey species, sheep have evolved to protect themselves from predators and increase their chance of survival by showing minimal responses to pain (Gougoulis, Kyriazakis & Fthenakis, 2010). This makes it challenging to assess the level of pain lambs may be experiencing following tail docking, particularly after the first hour as the display of acute behavioural indicators of pain reduces significantly (Molony, Kent & McKendrick, 2002). Specific behaviours have been validated as indicators of acute pain in lambs, including active pain avoidance behaviours, abnormal body postures and grimace scales (Grant, 2004; Guesgen et al., 2016; Kent, Molony & Graham, 1998; Marini, Colditz & Lee, 2021; Molony, Kent & McKendrick, 2002). However, these behavioural assessments still do not provide a direct measure of pain, nor the emotional state of the animal. Pain is an individual emotional experience and measures of affective state may complement the observation of pain-specific behaviours, to determine how that animal perceives a painful experience. Emotional reactivity tests are often used to determine an animal’s affective state, particularly after experiencing a stressor (Doyle et al., 2011; Grillon, 2008; Marchewka et al., 2016). Measures of attention bias can also indicate negative affect (Monk et al., 2020) and it has been reported that castrated lambs are more likely to show attention to the surrounding environment than non-castrated lambs (Cho, Lee & Small, 2020). The use of an attention bias or reactivity test in lambs following tail docking may give some insight into the emotional state of lambs in pain. Such insight could provide knowledge on their experience of pain in relation to duration and intensity resulting in improved assessment of pain and pain relief treatments. Emotional reactivity and response to stress has also been reported to differ in sheep depending on their sex and breed (Vandenheede & Bouissou, 1993) and therefore we might expect to see variation in response between male and female lambs.

The aim of the current study was to assess sex differences in pain using behavioural responses and an emotional reactivity test as a measure of emotional state. Two hypotheses were tested. The first: Female lambs undergoing tail docking would be more sensitive to the pain than male lambs. Increased sensitivity to pain would constitute displaying an increase in known behavioural indicators of pain following ring tail-docking. The second hypothesis: Female lambs in pain would show a higher state of emotional reactivity in the test compared to male lambs in pain and to lambs not in pain. A higher state of emotional reactivity would constitute an increased display in fearful behaviours and a decrease in exploratory behaviours.

Materials & Methods

All animal experimentation and modifications to protocol were approved by the CSIRO Armidale animal ethics committee under the NSW Animal Research Act, 1985 (ref ARA 20/18). The study used a blinded controlled randomized block design, incorporating detailed individual behavioural pain indicators and emotional reactivity testing. The experimental unit was the individual animal. A pilot study was conducted to establish a protocol for attention bias testing in young lambs as a measure of emotional state (Lee et al., 2016; Monk et al., 2018), which was ultimately replaced by an emotional reactivity test in the main study (Erhard et al., 2004).

Animals

In total, 96 single-born Merino lambs between 6–10 weeks of age were used for this study (pilot and main study), with an average bodyweight of 15.87 ± 0.70 kg (average ± SEM) at the time of testing. All animals were obtained from the CSIRO Chiswick farm, Armidale, New South Wales, Australia. When not part of the experimental protocol, ewe-lamb pairs were kept as a flock in a paddock assigned by the farm manager and managed under standard husbandry practice. When the lambs were required, they were transferred with their mothers into an animal housing facility in six cohorts of 16 ewe-lamb pairs/cohort. The first cohort was used in the pilot study, while the other five cohorts were used in the main study. Lambs were penned in groups containing two male and two female lambs with their respective mothers. Grouping was based on bodyweight rankings, with pen allocation assigned sequentially by bodyweight within lamb sex. Lambs were marked with symbols applied to the wool using coloured spray (Colourflow, Heiniger Australia, Western Australia) for identification for observation of behaviour through video recording. Identification marks were randomised across treatments within a pen. Pen-side health inspections were carried out assessing movement and demeanour, immediately prior to entry into the animal housing facility to confirm acceptance into the trial. Each ewe-lamb pair was fed 800 g sheep pellets and 200 g chaff daily, water was provided ad libitum. Animals were kept in the animal housing facility for a maximum of 7 days. After all lambs had completed the reactivity test the research trial was concluded. Male lambs were then castrated and Sham lambs were tail docked with the Numnuts® device (Senesino, Brisbane, Australia) with NumOcaine™ (1.5 ml Lignocaine; Senesino, Brisbane, Australia) and all lambs received Buccalgesic OTM (10 mg/ml Meloxicam; Troy Laboratories, Glendenning, Australia) as per standard husbandry and were returned to the CSIRO Chiswick farm.

Emotional reactivity test arena and equipment

Attention bias and emotional reactivity testing for the pilot trial and the main experiment occurred in a 2.5 ×2.5 m arena with 2 m high opaque walls, located within the animal house. The arena contained a food bowl positioned between removable clear PVC barriers, to block physical access to the bowl from the side without blocking vision (Fig. 1). On the opposite wall to the food bowl was the location of the negative stimulus. Lambs entered the arena from a guillotine door on the left-hand side of the arena.

Figure 1 Schematic diagram of the test arena used during the pilot trial.

The diagram indicates the location of the food, entry door and threat door. Camera boxes were positioned in all corners of the arena, but only the bottom left corner box contained a GoPro camera (shaded grey).

Lambs were video recorded during habituation and behavioural testing using GoPro cameras (Hero5; GoPro, San Mateo, CA, USA) mounted on the rafters above the test arena as well as a camera mounted inside a shelf located at lamb height in the left-hand corner between the guillotine door and negative stimulus location. Identical shelves were located at the same height in the remaining corners of the arena.

Pilot study

A total of 16 ewe-lamb pairs were used in the pilot study. In the first week of housing, lambs were habituated to the testing arena with no negative stimulus present, to become familiar with and consume feed within the arena. Habituation was conducted daily and involved removing lambs from their pens and moving them to the test arena (2.5 × 2.5 m), for up to 30 min before being returned to their ewes. Inside the arena, lambs were presented with four brightly coloured bowls that contained feed pellets. The bowls were positioned between removable clear PVC barriers. Across the week, lambs underwent three habituation sessions of up to 30 mins in groups of four. Lambs then underwent two sessions of up to 30 mins in randomly assigned pairs (whichever lambs were caught first). Finally, the lambs underwent two, five-minute habituation sessions individually totalling seven habituation sessions/lamb.

After the habituation stage, a negative stimulus was trialled. All lambs underwent a baseline attention bias assessment using the test configuration displayed in Fig. 1, with no threat present. Each individual lamb was placed in the arena for 5 mins while behaviours were recorded live via feed from GoPro cameras. The following day, the arena was arranged so that a photograph of a dog was positioned on the wall opposite the positive food stimulus at the beginning of the test. An individual lamb entered the arena, and the following behaviours were recorded to indicate a fearful response to the threatening stimulus compared to baseline behaviour: duration looking at the dog (s), freezing responses (yes/no), frequency of entering the zones near the dog’s location, number of high and low vocalisations, latency to eat and quantity of food eaten (g). It was found that the photograph was inadequate as a negative stimulus for the first five lambs. Subsequent lambs were therefore tested with a live dog used as a threatening stimulus. The dog’s location was that of the photograph and the same behaviours indicating a fearful response were recorded.

During the test with a threat present, lambs did not consistently direct their attention towards the photograph of a dog or the live dog. Lambs also did not consistently display a fear response towards the photograph of a dog or to the live dog. It therefore appeared that the dog was not an innately threatening stimulus for the young lambs used in the pilot study. The proposed method used in the pilot trial for assessing attention bias was determined not to be suitable for use with young lambs and an alternative test of emotional reactivity including an alternative assessment of attention bias (see Emotional reactivity test) was used for the main study (Erhard et al., 2004).

Main study

The study was conducted across five cohorts of 16 lambs per cohort. Within each cohort, the lambs were ranked according to weight and sequentially blocked into blocks of four. Weighted blocks were rotated between pens each week. Lambs were randomly allocated, by picking coloured marbles from an opaque bag, from within each block to each treatment;

1) FRing–Female lamb, tail docked using standard elastrator pliers without analgesia

2) MRing–Male lamb, tail docked using standard elastrator pliers without analgesia

3) FSham–Female lamb, tail manipulated to mimic ring tail docking

4) MSham–Male lamb, tail manipulated to mimic ring tail docking

This provided 20 replicates per treatment, which would allow us to define the probability distribution around the mean counts with reasonable confidence to compare those 95% CI brackets. The number of animals required per group was based on a sample size determination calculation, using anticipated mean and standard deviation values obtained from previous tail docking and attention bias studies.

During treatment application, lambs were picked at random (whichever lamb was caught first). All lambs were placed on their back in a lamb marking cradle for treatment application. Ring lambs had an elastrator ring applied to the tail at the level of the third palpable joint according to standard industry procedures (Lloyd & Playford, 2013). After treatment, lambs were placed back in their pen for behavioural observation. During the main study, lambs were not given analgesia as the aim was to compare response to pain between male and female lambs and to measure the emotional reactivity of lambs in pain. These methods comply with The Australian Animal Welfare Standards and Guidelines for Sheep (AHA, 2014). Sham lambs were placed in the lamb marking cradle and had their tail manipulated as though a ring were to be applied, but no ring was applied. All animals received analgesia (see Animals) following the end of the experiment as per the CSIRO Chiswick farm’s standard husbandry procedures before being returned to the paddock.

Behavioural assessment

Video cameras were used to continuously record the behaviour of lambs in the study. For each pen, one camera (Dahua IR Dome 4 MP network camera: DH-IPC-HDBW2431RP-ZAS-27135-S2; Zhejiang Dahua Vision Technology Co., Ltd, Hangzhou, China) was mounted on roofing rafters at each end of the pen. Each camera provided a view of the entire area available to the lambs. The cameras were connected to digital video recorders and captured by SmartGuard software from Pacific Communications (PACOM), Macquarie Park, New South Wales, Australia. The assessment of the behaviour post-treatment focused on the acute pain related behaviours and postural behaviours associated with ring tail docking. The pain related behaviour assessment was conducted from video footage, the person assessing behaviour was a female research officer with previous training and experience in assessment of pain-related behaviours and was blinded to lamb treatment application. Assessment took place for one full minute at five-minute intervals for the first 45 mins post-procedure, according to the ethogram shown in Table 1. Postural behaviours recorded were normal upright, abnormal upright, unknown upright, normal lying, abnormal lying, and unknown lying (Table 1). The most prominent posture displayed in the one-minute observation period was recorded. Acute pain behaviour was assessed every five minutes for one full minute and a count of all behaviours observed in that minute was taken. The ethogram used in this study was based on validated behaviour patterns (Molony, Kent & McKendrick, 2002) described in previous studies (Grant, 2004; Kent, Molony & Graham, 1998; Marini, Colditz & Lee, 2021) of behavioural responses of lambs following ring tail docking. Grimace scales were not applied due to their limited validity in lambs (Evangelista, Monteiro & Steagall, 2022) and the woolly faces of the Merino lambs used preventing accurate classifications.

Table 1 Description of active pain avoidance behaviours and postures recorded every 5 min, for 1 min during the 45 mins post tail-docking.

The most prominent posture displayed in the one-minute observation period was recorded. For acute behaviour a count of all behaviours observed in that minute was taken.

Acute behaviour 	Abbreviation 	Description 	
Restlessness 	Rst 	Number of times lamb stood up and laid down. Instances of lamb rising as far as its knees included in the one count. 	
Foot stamping/kicking 	Fsk 	Either a front or hind limb (usually hind limb) was lifted and forcefully placed on the ground while standing or was used to kick while standing or lying. 	
Rolling 	Rl 	Rolled from lying on one side to the other without getting up. Half rolls where the lamb rolled on its back and then returned to lying on the same side included. 	
Jumping 	Jmp 	All four feet off ground simultaneously.	
Licking/biting wound site 	Lbw 	Movement of the head beyond the shoulder, including both looking and touching at the source of pain and grooming. 	
Head shake 	Sh 	Forceful voluntary shake of the head.	
Easing quarters 	Eq 	Abnormally lowers rear quarters (standing) or attempts to keep quarters off the ground (lying). 	
Tail wiggle	Tw	Lamb moves their tail about in a wiggling motion.	
Unsteady hindquarters	Uh	Lamb staggers or totters in hindquarters sideways.	
Postures 	Abbreviation 	Description 	
Normal upright	NU	Standing, walking or playing while exhibiting a usual posture or gait; smooth movements.	
Abnormal upright	AU	Standing exhibiting unusual posture e.g., rounded, hunched appearance; standing stretched; ataxia; jerky movements; walking unsteadily, backwards, on knees.	
Unknown upright	UU	Lamb is upright but cannot be seen clearly enough to distinguish type of upright.	
Normal lying	NL	Ventral recumbency, all legs tucked under body or very close to body.	
Abnormal lying	AL	Twisted lying; ventral recumbency with forelimbs tucked under body, one or both hind limbs partially or fully extended; including dog sitting and lateral lying (lateral recumbency with one shoulder on ground, hind limbs and/or forelimbs fully extended).	
Unknown lying	UL	Lamb is lying but cannot be seen clearly enough to distinguish type of lying.	

Emotional reactivity test

Following induction to the animal housing facility, lambs and ewes were given one day to habituate to the pens without additional disturbance. During the following two days lambs were habituated to the emotional reactivity test arena in groups of four. Lambs were kept in the arena for 10 mins, twice daily, to habituate them to the novel environment and brightly coloured feed bowls containing feed pellets.

One hour after treatment, lambs were individually moved into the test arena, in their treatment order, for behavioural testing. The behavioural test was divided into several consecutive phases: ‘Isolation’, ‘Novelty’, ‘Startle’. For the Isolation phase, individual lambs were moved into the arena; lambs were tested for 90 s without any threat present. After 90 s of isolation had passed, a novel object (brightly coloured umbrella) was introduced into the arena to begin the Novelty phase. The closed umbrella was placed through a hole in the arena on the opposite side to the food bowl location (Fig. 2), and held there until the lamb approached within approximately 20 cm. Once the lamb approached the closed umbrella, the umbrella was opened, the Novelty phase was ceased, and the next phase began. If the lamb didn’t approach the umbrella within 3 mins, the test was ceased, and the lamb was removed from the arena and returned to its pen.

If the tested lamb approached the closed umbrella and the umbrella opened, the Startle phase of the test began immediately. The lambs’ immediate startle responses were recorded, then the lamb was kept in the arena with the open umbrella for 90 s to assess their behaviour as a potential measure of attention bias. During this time, lambs were recorded for their latency to eat as well as attention towards the umbrella, assessed as duration looking towards the umbrella within the field of binocular vision. After 90 s the lamb was removed from the arena and returned to its pen.

Throughout all phases of the emotional reactivity test, the lamb’s behaviour was recorded using GoPro cameras as described in the ‘Emotional reactivity test arena and equipment’ section. Footage from the cameras was used to undertake behaviour assessments using The Observer XT software (Noldus, The Netherlands). The footage for two lambs were missing due to technical errors (one Ring Male and one Ring Female). Data from these animals were excluded from the analysis of emotional reactivity. The person performing the video observations post-treatment was blinded to treatment. Live pen side observations were also recorded for latency to eat and number of bleats during each phase of the test, as well as latency to approach the umbrella. The behaviours recorded during each phase of testing are defined in Table 2.

Figure 2 The final setup of the test arena for the emotional reactivity test.

(A) Schematic diagram and (B) photograph of the test arena used during the main study, indicating the location of the food, entry door and umbrella. The photograph depicts a sheep in the Startle phase of testing. Camera boxes were positioned in all corners of the arena, but only the bottom left corner box (shaded grey in panel (A)) contained a GoPro camera. Lines and numbers indicate the zones (1 –5) of the arena used during behavioural assessment. Zone 5 was only counted once the umbrella was introduced (Novelty and Startle phase).

Table 2 Behavioural measures recorded during the phases of the emotional reactivity test; Isolation, Novelty and Startle.

The phase(s) of the test in which a given behaviour was recorded is indicated by “Y”.

Behaviour	Definition	Isolation	Novelty	Startle	
Vigilance	Percentage of test phase duration spent with the head at or above shoulder height (%).	Y	Y	Y	
Feeding	Percentage of test phase duration eating food (%) and latency to eat from start of test phase (s).	Y	Y	Y	
Zones crossed	Number of zones crossed with both front feet (n).	Y	Y	Y	
Vocalisations	Number of open-mouthed and closed mouthed bleats recorded separately (n).	Y	Y	Y	
Sniff environment	Number of sniffs of the walls, floors or camera boxes (n).	Y	Y	Y	
Escape attempts	Number of escape attempts characterised by jumping or rearing towards wall (n).	Y	Y	Y	
Umbrella approach	Latency to approach umbrella (s).		Y		
Umbrella reaction	Categorical score of reaction to the umbrella entering or opening; None (no or minimal response e.g., head turn), Low (noticeable startle, steps backwards or moves a short distance while remaining within one zone), Med (quickly moves away but remains within the one zone or walks to another zone) or High (rapidly moves away across multiple zones, may attempt to escape). A jump with all feet off the ground was also scored separately (0/1).		Y	Y	
Umbrella attention	Percentage of test phase duration spent with the head oriented directly towards the umbrella (%).		Y	Y	

Statistics

Statistical analysis was conducted using R and R studio software (R version 4.0.2 “Taking Off Again”) (R Team, 2022). Data were tested for normality through visual inspection of residual plots and the Shapiro–Wilks test. A P-value of <0.05 was considered statistically significant and 0.1 >P ≥ 0.05 was considered a statistical tendency. Data are presented as means ± standard error of the mean unless otherwise stated. There were no data exclusions for the analysis of pain behaviours.

Pain behaviours

The packages lme4 (Bates et al., 2015), ggpubr (Kassambara, 2020), dplyr (Wickham et al., 2019) and car (Fox & Weisberg, 2018) were used. The frequency of occurrence for individual postures was too low for data analysis, therefore categories were combined as upright (normal upright, abnormal upright, unknown upright), lying (normal lying, abnormal lying, unknown lying) and abnormal postures (abnormal upright, abnormal lying). Lying and abnormal postural behaviours were analysed using a non-linear mixed effects model, fixed factors included Treatment (n = 20 per group), Sex, Cohort, Pen and the interaction of Treatment × Sex, with Lamb included as a random effect. Sum of all abnormal postures was not normally distributed and was analysed using a Generalized linear mixed model with Poisson distribution, fixed effects and the random effect fitted were the same as other postures. The acute pain behaviour categories were combined for analysis. Acute pain behaviour counts were analysed using a Generalized linear mixed model with a Quasi-Poisson distribution due to over dispersal. Fixed effects included Treatment (n = 20 per group), Sex, Cohort, Pen and Time as well as the interaction of Treatment × Sex, with lamb included as a random effect.

Emotional reactivity

Statistical analyses were divided by test phase. All animals completed the Isolation test phase and were given an opportunity to approach the umbrella in the Novelty phase. Latency to approach the umbrella was analysed using a Cox proportional hazards model with the survival (Therneau, 2022; Therneau & Grambsch, 2000) package, fitting Sex and Treatment as fixed effects. Latencies were deemed as censored results if the lamb failed to approach the umbrella within 180 s. Reactions to the closed umbrella entering the test arena and to the umbrella opening were analysed using Fisher’s Exact Tests. The remaining Novelty test data analysis included only those that did not approach the umbrella within 180 s (n = 30), while only lambs that approached the umbrella moved on to the Startle phase (n = 48). Reactions to the umbrella opening in the Startle phase were analysed using Fisher’s Exact Tests.

Principal component analysis (PCA) was used to condense correlated behavioural measures within each phase of testing into their principal components, using the psych (Revelle, 2022) package. Variables were normalised using the scale function in R, then the scaled variables were checked for multicollinearity by analysing the correlation matrix. Variables with any correlations above 0.90, or with most correlations below 0.30, were excluded from the analysis (Field, Miles & Field, 2012). Sampling adequacy was confirmed by conducting Bartlett’s sphericity tests (P < 0.05) and calculating Kaiser-Meyer-Olkin (KMO) measures of sampling adequacy. The KMO statistic varies between 0 and 1, where a value close to 1 is considered ideal, while variables with a score < 0.5 were deemed unacceptable (Field, Miles & Field, 2012). Factors included in the analyses were subjected to varimax rotation. The number of rotated components used in further analyses were selected based on the number needed to explain more than 75% of the total variance and by checking the PCA model residuals (less than half with absolute values > 0.05), fit (> 0.9) and communality (all communalities > 0.7).

The selected rotated components within each test phase were then used to determine the effect of Sex and Treatment on the independent dimensions underlying the lambs’ behavioural reactivity. The rotated components were analysed using linear mixed effects models with the package nlme (Pinheiro & Bates, 2022; Pinheiro & Bates, 2000), fitting fixed effects of Treatment, Sex, Cohort, Pen and the interaction of Treatment × Sex and a random effect of Lamb (Field, Miles & Field, 2012). All of the rotated components for the Isolation phase were transformed to maintain the assumption of normality of the model residuals. A log transformation was applied after adding an integer of 2.7, 2.6 and 2.3 for the 3 rotated components respectively, such that the minimum value for each rotated component was ≈ 1. The second rotated component for the Novelty phase and the third rotated component for the Startle phase were log transformed to ensure normality of model residuals after adding integers of 2.4 and 2.1 respectively. Interactions and fixed factors were removed from the models using a backward elimination approach, considering the lowest Akaike Information Criterion (AIC) as an indicator of model fit.

Results

Pain behaviours

Only the results of mean sum of abnormal postural behaviour and lying behaviour are presented, as upright behaviour is the opposite of lying and all abnormal postures are mutually exclusive. For lying behaviours there was a treatment (F1,69 = 19.0), cohort (F4,69 = 7.2), and pen (F3,69 = 3.7) effect observed (all P < 0.05), but no effect of the interaction of treatment x sex (F1,69 = 0.5, P = 0.47). Lambs in the Sham group were observed lying fewer times than the Ring lambs (Fig. 3). Lambs in cohort 2 were observed lying more than cohort 1 (t 69 = 2.3, Mean count = 5.2 ± 0.5 vs 4 ± 0.5 respectively, P = 0.02). Cohort 5 lambs lay down fewer times than cohort 1 (t 69 = −2.8, Mean count = 2.5 ± 0.5 vs 4 ± 0.5 respectively, P = 0.005).

Figure 3 Mean count of all lying behaviours observed for male and female lambs within the Sham and Ring treatment groups in the 45 min post tail docking.

There was a difference in behaviour between the treatment groups (P = 0.01), there was no significant difference between sexes (P = 0.12) or sex within treatment (P = 0.46).

Ring lambs displayed more abnormal postures compared to Sham lambs (z = −3.8, P = .0001, Fig. 4). Each subsequent cohort (2 to 5) after cohort 1 either tended to display or displayed more abnormal postures following tail docking. Cohorts 2 and 3 displayed an additional mean of 1 ± 0.4 observed abnormal postures (z = 2.8 and 2.4, P < 0.05), and cohorts 4 and 5 displayed an additional mean of 0.7 ± 0.4 (z = 1.7 and 1.8, P < 0.08), observed abnormal postures compared to cohort 1 mean count of 0.5 ± 0.3. Pen influenced the behaviour display of lambs with animals in pen two and three displaying fewer observed abnormal postures compared to pen one, with a decrease in mean count of −0.6 ± 0.2 (z = −2.2, P = 0.02) and −0.7 ± 0.3 (z = −2.4, P = 0.01) respectively.

Figure 4 Raw data for mean count of all abnormal postural behaviours observed for male and female lambs within the Sham and Ring treatment groups in the 45 min post tail docking.

Acute pain behaviours were assessed every 5 mins for the 45-min period following treatment. There was an effect of cohort (χ2(4) = 119.3, P < 0.001), pen (χ2(3) = 7.9, P < 0.05) and time (χ2(8) = 678.3, P < 0.001) as well as sex (χ2(1) = 54.2, P < 0.001) and treatment (χ2(1) = 1450.3, P < 0.001) and an effect of their interaction (χ2(4) = 119.3, P < 0.05). There was a difference in overall mean count of acute behaviours displayed between male and female lambs that were tail docked (t = −4.9, P < 0.001, Fig. 5). Lambs in the Sham group displayed significantly fewer acute behaviours than the tail docked lambs (t = −7.2, P < 0.001). There were no differences in behaviour seen between sham male and female lambs (P = 0.5).

Figure 5 Raw data for mean count of all acute pain behaviours observed for male and female lambs within the Sham and Ring treatment groups in the 45 min post tail docking.

The presence of the asterisk indicates a significant difference within treatment. P values were derived from the Generalized linear mixed model.

Emotional reactivity

Isolation phase

In the Isolation test phase (n = 78), six variables fit the criteria for inclusion in a PCA. Sampling adequacy was verified as ‘good’ by the overall KMO (0.73) and values for individual variables ranging from 0.5 to 0.82. Bartlett’s test of sphericity indicated that correlations between items were sufficiently large to conduct a PCA (χ2(15) = 259, P < 0.001). Three components were retained for further analysis. Table 3 shows the factor loadings on each variable after rotation. Based on the items that cluster on each component, we labelled component 1 as fearful responses, component 2 as exploration and component 3 as a social response for easier interpretation, however other interpretations of these behaviour profiles should also be considered. For all 3 principal components, linear mixed effects models indicated no significant effect of treatment, sex or their interaction on behavioural reactivity. The AIC supported inclusion of cohort only in all models.

Table 3 Summary of the principal component analysis results for the Isolation phase of behavioural reactivity testing (n = 78).

Absolute factor loadings over 0.6 are written in bold.

	Varimax rotated factor loadings	
Isolation behaviour	Component 1
(fearful)	Component 2 (explorative)	Component 3
(social)	
Zones crossed	0.88	0.19	0.20	
High vocalisations	0.86	0.37	−0.11	
Vigilance %	0.62	0.66	0.22	
Latency to eat	0.43	0.74	0.19	
Sniff environment	0.17	0.89	0.10	
Low vocalisations	0.08	0.17	0.98	
Eigenvalues	2.10	1.97	1.10	
% of variance	35	33	18	

Novelty phase

For latency to approach the umbrella (n = 78), the Cox proportional hazards model fitting treatment and sex as fixed effects was not significant (Likelihood ratio test = 3.38, df = 2, P = 0.2). There was no effect of treatment group on reaction to the closed umbrella entering the arena (number of lambs with a High reaction in FRing = 7, MRing = 5, FSham = 7, MSham = 5; Fisher’s Exact Test, P = 0.905).

For the Novelty phase including only animals that failed to approach the umbrella (n = 30), five variables fit the criteria for inclusion in a PCA. Sampling adequacy was verified as ‘good’ by the overall KMO (0.71) and values for individual variables ranging from 0.5 to 0.86. Bartlett’s test of sphericity indicated that correlations between items were sufficiently large to conduct a PCA (χ2(10) = 80, P < 0.001). Three components were retained for further analysis. Table 4 shows the factor loadings on each variable after rotation. Based on the items that cluster on each component, we labelled component 1 as high activity fearful responses, component 2 as exploration and component 3 as an active response.

Table 4 Summary of the principal component analysis results for the Novelty phase of behavioural reactivity testing, for sheep that failed to approach the umbrella (n = 30).

Absolute factor loadings over 0.6 are written in bold.

	Varimax rotated factor loadings	
Novelty behaviour	Component 1
(fearful)	Component 2 (explorative)	Component 3
(active)	
Attention to umbrella	0.92	0.03	0.10	
Latency to eat	0.89	0.31	0.00	
Vigilance %	0.83	0.36	0.30	
Sniff environment	0.24	0.94	0.18	
Zones crossed	0.12	0.17	0.98	
Eigenvalues	2.41	1.14	1.09	
% of variance	48	23	22	

For the first rotated component, the AIC supported inclusion of a treatment by sex interaction in the model, however the interaction term was not significant (χ2(1) = 2.3, P = 0.13). There tended to be a treatment effect (χ2(1) = 3.0, P = 0.084), whereby Ring animals scored higher on the ‘fearful’ index than Sham animals (Fig. 6; Least squares means 0.30 ± 0.26 and −0.38 ± 0.26 respectively, averaging over the levels of sex). For the second rotated component, the AIC supported inclusion of sex in the model only, for which there tended to be a sex effect (χ2(1) = 3.7, P = 0.056), whereby Female animals scored higher on the ‘explorative’ index than Male animals (Least squares means 0.96 ± 0.12 and 0.67 ± 0.10 respectively). For the third rotated component, the AIC supported the null model, with no significant effect of treatment, sex or the treatment by sex interaction.

Figure 6 Raw mean ± s.e.m. of rotated principal component scores reflecting ‘fearfulness’ within the Novelty and Startle test phases for Sham and Ring treatment groups post tail docking.

P values were derived from linear mixed effects models fitting either sex, treatment and a treatment by sex interaction or treatment only, for each test phase respectively.

Startle phase

For the Startle phase including only animals that approached the umbrella (n = 48), there was no effect of treatment group on reaction to the umbrella opening (number of lambs with a High reaction in FRing = 3, MRing = 7, FSham = 6, MSham = 4; Fisher’s Exact Test, P = 0.564) or jump response to the umbrella opening (number of lambs that jumped in FRing = 5, MRing = 3, FSham = 4 MSham = 2; Fishers Exact Test, P = 0.777). Five variables fit the criteria for inclusion in a PCA. Sampling adequacy was considered mediocre, with an overall KMO of 0.61 and values for individual variables ranging from 0.5 to 0.64. Bartlett’s test of sphericity indicated the correlations between items were sufficiently large to conduct a PCA (χ2(10) = 33, P < 0.001). Three components were retained for further analysis. Table 5 shows the factor loadings on each variable after rotation. Based on the items that cluster on each component, component 1 was labelled as fearful responses, component 2 as an attentive response and component 3 as an escape response.

Table 5 Summary of the principal component analysis results for the Startle phase of behavioural reactivity testing, for sheep that approached the umbrella (n = 48).

Absolute factor loadings over 0.6 are written in bold.

	Varimax rotated factor loadings	
Startle behaviour	Component 1
(fearful)	Component 2
(attentive)	Component 3 (escape)	
Latency to eat	0.84	0.06	0.01	
Vigilance %	0.83	0.18	0.05	
High vocalisations	0.64	−0.54	0.16	
Attention to umbrella	0.20	0.89	−0.13	
Escape attempts	0.07	−0.14	0.99	
Eigenvalues	1.83	1.15	1.02	
% of variance	0.37	0.23	0.20	

For the first rotated component, the AIC supported inclusion of treatment only in the model. The effect of treatment was significant (χ2(1) = 5.6, P = 0.018), whereby Ring animals scored higher on the ‘fearful’ index than Sham animals (Least squares means 0.31 ± 0.19 and −0.33 ± 0.20 respectively). There was no effect of sex or the sex by treatment interaction on behavioural reactivity. For the second rotated component, the AIC supported the null model, with no significant effects of sex, treatment or the sex by treatment interaction. For the third rotated component, the AIC supported the model fitting cohort only, with no significant effects of sex, treatment or the sex by treatment interaction.

Discussion

Pain research in livestock often focuses on the impact that castration has on male animals, with limited research looking at the different impacts that painful husbandry procedures may have on female lambs compared to males. The aim of the current research was to examine how the sex of lambs may impact their experience of pain in response to the procedure of tail docking. During the 45 min after ring tail docking, female lambs that were tail docked displayed more acute pain related behaviours when compared to their male counterparts. This display of different behaviours between sexes was not seen in Sham group lambs. A difference in abnormal postural behaviour was also seen between Sham and Ring lambs however no sex differences were observed for these behaviours. An emotional reactivity test was applied as a measure of emotional state in the lambs. The Ring lambs displayed more fearful behaviours during the Novelty and Startle phases of the test compared to Sham lambs, but against our expectations, no sex effect was observed.

Previous research on the impact of painful husbandry procedures on lambs have included both sexes without direct comparison between them (Grant, 2004; Price & Nolan, 2001; Shutt et al., 1987; Small et al., 2020). In this study, the direct comparison of the tail docking procedure between the sexes found that female lambs displayed more pain avoidance behaviours in the hour following tail docking compared to the male lambs, indicating a higher sensitivity to the pain caused by the procedure. Lamb age is an important factor in the stress response caused by pain, with sex differences not being observed in very young lambs (1–7 days of age) (Guesgen et al., 2011; Mellor & Murray, 1989; Turner et al., 2006), but by 8 weeks of age female lambs’ cortisol response to tail docking is significantly greater than that of male lambs (Turner et al., 2006). Lambs in Australia commonly undergo painful husbandry procedures between 4–12 weeks of age, therefore, sex is likely to impact the pain response of the lambs in commercial settings. This also has implications for experimental testing of new pain-relief pharmaceuticals to ensure they are validated as effective for both sexes.

Lambs undergoing ring tail docking were observed lying down more often than the sham lambs, which is consistent with reporting of behaviour in other studies following ring tail docking (Lester et al., 1996; Molony, Kent & Robertson, 1993; Small et al., 2020). The observation of increased lying is indicative of the discomfort caused by the procedure and is specific to lambs undergoing ring castration and tail docking. Lying behaviour reduces in ring castrated and tail docked lambs when they are provided pain relief, whereas surgically castrated and tail docked lambs tend to display standing behaviours as a sign of discomfort (Small et al., 2014). There is also growing evidence that the pain of tail docking and castration procedures lasts beyond current commercially available analgesic coverage (Mellor & Stafford, 2000) and potentially beyond wound healing (Johnston et al., 2022), with neuroma formation reported in docked lambs’ tails (French & Morgan, 1992; Larrondo et al., 2019). The presence of neuromas can lead to persistent pain after the tail wound is healed, however, it is difficult to assess the chronic pain that painful husbandry procedures may cause in lambs (Mellema et al., 2006). The lambs tail docked in this study displayed behaviours indicative of acute pain and are therefore at risk of this pain developing into chronic pain or from suffering from allodynia and hyperalgesia (Johnston et al., 2022; Viñuela Fernández et al., 2007). There is also evidence of sex differences on the neuroimmune mediation and subsequent chronification of pain, with females more likely to develop chronic pain (as reviewed by Midavaine et al., 2021; Mogil, 2020; Price & Ray, 2019). As well as evidence of discrepancy of analgesic efficacy between sexes, with females not responding to analgesia as well as males (Mogil, 2020; Price & Ray, 2019). This has implications for the ewes that are retained long term within the sheep industry. If female lambs are more sensitive to pain and the current analgesic options available are potentially inadequate, there may be ewes in the national flock suffering from undetected and untreated chronic pain, thus negatively impacting their welfare. Furthermore, as female sheep reach maturity, their stress responses and thus affective experience of pain may vary across their oestrus cycle (Komesaroff et al., 1998), an additional factor to consider for any necessary flock pain management.

The emotional reactivity tests were able to detect some differences between the treatment groups, with Ring animals more likely to show putatively fearful behaviours than Sham animals, particularly during the Startle phase, but there were minimal effects of sex observed. Previous tests of emotional reactivity and fear in lambs and adult sheep have found some sex differences of varying magnitudes (Atkinson et al., 2022; Boissy et al., 2005; Roussel et al., 2008; Vandenheede & Bouissou, 1993; Viérin & Bouissou, 2003). This suggests the emotional reactivity of males and females does differ, but is likely to be dependent on age, breed, and the exact behavioural reaction being assessed. The lack of sex or ‘sex within treatment’ effect in the current study contrasted with the sex differences seen in the display of acute pain behaviours. The females may have been experiencing more pain, but the novelty of the testing situation may have been distracting, directing attention away from pain (Bushnell, Čeko & Low, 2013) to their new surroundings. This suggestion is supported by the female’ tendency to sniff their environment more in the Novelty subtest compared to males.

The observation of the Ring lambs showing more behaviours associated with fearfulness than Sham lambs is consistent with our predictions of greater emotional reactivity in animals experiencing pain. Sheep are prey animals where their stoical behaviour can often prevent overt displays of pain and distress (Gougoulis, Kyriazakis & Fthenakis, 2010) but this doesn’t preclude the experience of pain. Emotional reactivity tests provide further insight into the affective state of the sheep, where the tail docking in this study led to greater fear-related behaviour consistently across both the Novelty and Startle phases. During the Startle phase, Ring animals showed greater vigilance and a longer latency to eat indicative of higher arousal, and more vocalisations suggesting a negative valence. A negative state while experiencing pain has also been demonstrated in laying hens suffering from keel bone damage (Armstrong et al., 2020; Wei et al., 2022) and in dairy calves following disbudding (Neave et al., 2013). In the Novelty phase, Ring animals tended to score higher on a ‘fearful’ score comprising high loadings of attention to the umbrella, latency to eat and vigilance. While these were interpreted as representing a fearful response, they might also be considered as increased interest or curiosity in the novel object. Hypervigilance while in pain may also have evolutionary advantages, where observations in squid (Crook et al., 2014) and mice (Lister et al., 2020) showed that induced pain resulted in a reduced predation risk or increased attention to a perceived predator threat respectively. Despite observing an effect of pain on emotional reactivity, pain did not appear to impact startle or jump responses to the umbrella entering the arena or opening. This contrasts with previous studies in sheep and other species that have demonstrated the startle response can be potentiated by negative affect (Doyle et al., 2015; Grillon, 2008).

The emotional reactivity test sequence employed in this study was limited by requiring approach responses from animals to proceed through the entire testing sequence. Thirty out of the 78 lambs did not approach the novel object (umbrella), preventing recording of responses during the last phase of testing. Comparable limitations have been observed in previous applications of the test sequence (Erhard et al., 2004) or similar test sequences (Roussel et al., 2008) where not all individuals will complete the full testing regime restricting the dataset, and, thus, losing potentially informative animals. There are many behavioural tests that have been validated for measuring emotional reactivity in sheep, assessing different aspects of temperament (reviewed in Dodd et al., 2012). It is important to consider that tests eliciting the greatest fear/reactivity or test situations that have additive effects (e.g., a novel object presented in a novel environment) may lead to floor effects and reduced sample sizes. Additionally, prior experiences of the animals occurring in close temporal association, such as handling and application of treatment procedures, may compound and further affect specific behavioural test outcomes.

The attention bias test paradigm developed by Lee et al. (2016) presents a method for assessing affective state that does not require an approach response or the meeting of learning criterion, but this method was deemed unsuitable for use in young lambs during our pilot study, due to an inconsistent response to the predator threat stimuli. Porter & Bouissou (1999) found that 3–4-week-old lambs did not display differential avoidance of or attraction toward images of a ewe versus a dog, while adult ewes responded negatively to the dog image. Young (2006) found no differences in restlessness displayed by 1–6-week-old castrated lambs in the presence of a live dog compared to a cardboard box or live goat. Together, these findings suggest the young lambs may not have had enough experience with dogs to have made a negative association with them and different threatening stimuli should be used to assess attention bias in young animals. Alternatively, the Startle phase of the main study might itself be interpreted as an attention bias test, where the umbrella poses a potential threat to the lambs. Duration looking toward the threat is a key measure of attention bias (Crump, Arnott & Bethell, 2018) that loaded highly on the second principal component, for which no treatment effect was observed. On the other hand, general fearfulness was impacted by treatment in both the Startle and Novelty phases. This may indicate attention bias is not impacted by pain or lamb sex, or perhaps the Startle phase may be more correctly interpreted as a novel object test instead, considering that the umbrella is not innately threatening.

Conclusions

The current study aimed to assess sex differences in pain using behavioural responses and an emotional reactivity test as a measure of emotional state. Lambs that underwent tail docking displayed more behavioural indicators of pain, with female lambs displaying more pain avoidance behaviours than male lambs, indicating higher sensitivity. This study also demonstrated that lambs in pain are experiencing a negative affective state, with more tail docked lambs displaying behaviour associated with fearfulness during Novelty and Startle phases of the emotional reactivity test, compared to the control lambs. However, no sex differences were seen during the emotional reactivity test. Female sheep make up a majority of Australia’s flock and the implication of increased sensitivity to; and the potential chronification of pain should be investigated.

Supplemental Information

Supplemental Information 1 ARRIVE 2.0 checklist

Click here for additional data file.

We thank Troy Kalinowski, Tim Dyall and Akash Pratap for their assistance throughout the project. The project was conducted on Anēwan land, and the researchers would like to acknowledge the Anēwan people as the traditional custodians of the land as well as their enduring connection to it.

Additional Information and Declarations

Competing Interests

Author Contributions

Animal Ethics

Data Availability

The authors declare there are no competing interests.

Danila Marini conceived and designed the experiments performed the experiments, analyzed the data, prepared figures and/or tables, authored or reviewed drafts of the article, and approved the final draft.

Jessica E. Monk conceived and designed the experiments, performed the experiments, analyzed the data, prepared figures and/or tables, authored or reviewed drafts of the article, and approved the final draft.

Dana L.M. Campbell conceived and designed the experiments, performed the experiments, authored or reviewed drafts of the article, and approved the final draft.

Caroline Lee conceived and designed the experiments, authored or reviewed drafts of the article, and approved the final draft.

Sue Belson performed the experiments, authored or reviewed drafts of the article, and approved the final draft.

Alison Small conceived and designed the experiments, authored or reviewed drafts of the article, and approved the final draft.

The following information was supplied relating to ethical approvals (i.e., approving body and any reference numbers):

All animal experimentation and modifications to protocol were approved by the CSIRO Armidale animal ethics committee (ref ARA 20/18).

The following information was supplied regarding data availability:

The data is available at the CSIRO Data Access Portal: Marini, Danila; Monk, Jessica; Campbell, Dana; Lee, Caroline; Belson, Sue; Small, Ali (2022): Pain behaviours and emotional reactivity following tail-docking in male and female lambs.. v2. CSIRO. Data Collection. https://doi.org/10.25919/1z7b-ed57.

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
