# Peer review of "Sex impacts pain behaviour but not emotional reactivity of lambs following ring tail docking"

_PeerJ, doi:10.7717/peerj.15092_

## Round 0.1 · original submission · Major Revisions

Thank you for submitting this interesting study to PeerJ. I regret that I am unable to accept the manuscript for publication, at least in its present form. However, I am prepared to consider a new version that carefully takes into account the suggested revisions. The reviewers liked many aspects of your study but also highlighted some important parts that require soem changes. These need to be addressed in detail in a new version. Such a revised manuscript is likely to be reviewed again and there is no guarantee of acceptance. When you revise the study, please prepare a detailed explanation of how you have dealt with all the reviewer comments. Please note that Reviewer 2 submitted their review as a separate file.

·

Basic reporting

The paper is well structured and written in clear and professional English, with sufficient references, barring some minor grammatical issues that require correction:

Line 65 : “to reducing the risk” should be corrected to either “to reduce the risk” or “reducing the risk”

Line 76 - 77 : There appears to have been an error in the phrasing of the sentence that may need correcting for clarity/correctness - “...has observed age by sex differences in the response to pain.”

Line 199: I believe there is a missing “the” between “based on” and “sample size determination calculation”

Line 202: Some revision needed to ensure correct English. My suggestion would be the inclusion of “were” between “lambs” and “picked” and the ending of the sentence after the bracketed statement and capitalisation of the following “They”.

Line 206: “measure of” should be “measure the” or alternatively “to provide a measure of the”

Line 232 – 234: The sentence beginning “Lambs were…” needs breaking into smaller statements for ease of reading.

Line 275: Potential accidental use of “abnormal” to describe the “normal” group assuming my understanding of which groups were analysed using which model methods is correct.

Line 343: There appears to be an additional “a” in “as well as a sex”

During the introductory discussion of current methods of behavioural assessment of pain in lambs (Lines 87-101) it is mentioned that current methods are insufficient as they “do not provide a direct measure of the emotional state of the animal.” A short, referenced expansion on this point to better establish the current methods for an unfamiliar reader may be useful here. Additionally, such an inclusion could better lead into the discussion of, and assist in establishing the potential of, the emotional reactivity assessment to supplement these behavioural assessments.

From Line 94 to Line 101 the authors describe the usefulness of a method of understanding affective state for informing husbandry practices and list two examples. The current structure of this section may read as disparate ideas despite making a valuable point and giving a foundational outline of the logic in the use of emotional reactivity tests. I would suggest some minor revision of these sentences to better tie them together and lead to the final sentence point that such methods provide valuable insight.

The data collected are readily available from the supplied repository link and are clearly labelled and laid out.

The box plots and bar charts included are clear with appropriate legends, axis labels and figure captions.

Experimental design

For the final introductory paragraph (Lines 103 – 107): The aims and research questions of the study are clearly stated, and relevant. The meaningfulness of the research has by this point been well established throughout the introduction. Additionally, the authors may consider the inclusion of an explanation of what would constitute as being “more sensitive to the pain” at this stage. Similarly, an expansion of what shows “a higher state of emotional reactivity” may also be useful context at this stage of the introduction despite being made clear later in the paper.

The methods are clearly outlined and include a detailed outline of performed pilot work. Figures and descriptions of the experimental setup are clear and concise, while also detailing specific suppliers, makes, and models of equipment used, allowing for ease of replication. The full process of selection, group allocation, and habituation are clearly described. The inclusion of this pilot work and the resulting changes are invaluable in understanding the decision-making processes resulting in the final methodology moving from attention bias to emotional reactivity.

On line 199 the authors report the use of sample size determination calculations to ensure comparisons are valid. Specifics of the calculation(s) used, and their outcome, would be helpful in confirming this.

The ethograms included are clear and detailed allowing for the easy replication of observations. Postural counts may need some further elaboration as to how they were assigned to the entire observed minute – if a lamb was observed in multiple postures across the 1 minute observational period was only the most prominent recorded? Additionally, readers may find it useful for a short justification of the use of postural counts rather than a record of postural duration, or perhaps the recording the occurrence of transitional behaviours between postures. The current methodology may neglect important finer grain details as to the stochasticity of postural state across the 1 minute observation periods or the entire 45 minute period which could potentially be meaningful to the research aims.

Diagrams of the pilot and main study experimental setups are clear and allow ease of replication. One addition that could be made relates to the use of “zones” within the ethogram behavioural descriptions. An indication of the borders and number of these zones within the diagrams would be useful.

The clarity in reporting of methodology continues during the explanation of statistical analyses with all software and packages reported including version numbers and references. Details are given on the models used and the assignment of factors as either fixed or random within the models. The inclusion of model formulae could help to present this. Additionally, a brief reasoning for the assignment of effects as either fixed or random may be of interest to readers as some categorical effects included here as fixed effects (namely cohort and pen) could instead be included in models as additional random effects (beyond the individual difference random effect of lamb).

From line 286 onwards the number of animals is reported as n=78 when previously this was 80. Clarification on the missing 2 animals would be useful, or a response identifying the line where the withdrawal of these 2 is reported, would be appreciated.

The paragraph from line 297 to 307 describes the process of applying PCA to the behavioural measures with details of the checks performed (KMO and Bartlett’s Sphericity). Inclusion of thresholds used for exclusion would be helpful to those that may not be familiar with the tests or have differing assumptions of acceptability prior to factor analysis. Additionally, the exclusion of behaviours with poor between animal variability is mentioned and details of the thresholds used for decision making would be valuable context. Further details of the scaling performed prior to the assessment of multicollinearity may also be useful for potential replication.

The provided ethics documentation and details included within the methodology are more than sufficient to support the conformity of this work to ethical standards and the elicitation of negative affect in the lambs only as necessary. Lambs were kept with their ewe except during short experimental or habituation sessions minimising separation anxiety related stressors. Additionally, the withholding of analgesia during and immediately following the ringing of tails is justified as replicating the Australian requirements and necessary for the study. Suffering was minimised as analgesia was provided following standard husbandry practice once the observational period had finished and prior to the lambs return to paddock.

Validity of the findings

Results are reported clearly and concisely. One extremely minor comment is the description of postural events which could be further clarified to reflect the recording of observed counts rather than exact counts or duration. Additionally, the reporting of a lack of difference in groups seen on Line 348 could include test statistics for the sake of completion.

Components selected from the PCA are assigned names relating to the features that cluster closest to them. This is useful in conceptualising the components but could be potentially misleading if assumptions are incorrect and this could be addressed within the text. For example, behaviours such as “Sniff environment”, or “latency to eat” are taken to be indicative of exploratory behaviour in the isolation phase, which is possibly true, but may also be displacement behaviours, indicators of heightened arousal, or of any number of unknowns. The text should make it clear that these names are educated assumptions or the components could be renamed to be less prescriptive.

The discussions of the results are in depth and provide valuable context based on previous research. However, one paragraph (Line 439 to 459) may need revising to better incorporate the valuable points made regarding neuroma development post tail docking and effects of sex differences on the chronification of pain to the studies results. The implications of these points are relevant and important but more direct linkage to the findings of the study would be beneficial to the discussion. Additionally, the potential implications of these points may be suited for inclusion within the introduction as support for the importance of this and future research in the area.

The conclusions summarise the research effectively and refer back to and address the aims and research questions initially laid out in the introduction.

·

Basic reporting

Please see attached document.

Experimental design

Please see attached document.

Validity of the findings

Please see attached document.

Additional comments

Please see attached document.

Reviewer 3 ·

Basic reporting

no comment

Experimental design

1 - It is discussed that females are more sensitive to tail docking, but the authors do not show female estrous cycles. several studies have been discussing that the sensitivity of females is associated with alteration, it would be interesting for the authors to discuss or analyze this variable in the study.

2 - As the test was recorded, I suggest analyzing the freezing behavior of the animals, this would help to corroborate the discussion regarding the issue of fear in animals.

3 - The authors argue that "Despite observing an effect of pain on emotional reactivity, pain did not appear to startle or jump responses to an umbrella entering the arena or opening it. This contrasts with previous studies in sheep and other species that demonstrated that the response of startle may be potentiated by negative affect (Doyle et al. 2015; Grillon 2008)", I think it would be interesting to discuss resilience to stress and the estrous cycle in females.

Validity of the findings

The current study aimed to assess sex differences in pain using behavioral responses and an emotional reactivity test as a measure of emotional state. It has great regional implications and brings important discussions about the quality of management of sheep herds and animal welfare.

Additional comments

no comment

---

## Round 0.2 · accepted · Accept

This is a really interesting and thorough study of pain perception in lambs/sheep.

·

Basic reporting

Following re-review of the manuscript, alongside the recommendations of myself and other reviewers, I am satisfied that the reporting of the research is suitable for publication. The article is well written, structured and areas highlighted as lacking in clarity have been refined satisfactorily.

Experimental design

I am also satisfied that the description of aims, research questions, and methods have been clarified suitably. The adjustments made to the manuscript serve to reduce ambiguousness present in some areas which was highlighted previously by myself and other reviewers.

Validity of the findings

The clarifications and additions made in response to review comments are appreciated and better outline the assumptions and decisions made in the analyses and their interpretation.

Additional comments

The authors have thoroughly addressed all reveiwers comments to a satisfying degree. Where suggestions have not been incorporated a detailed and sufficient rebuttal has been given. I believe the paper outcomes and overall clarity are much improved and suitable for publication.

·

Basic reporting

No comment

Experimental design

No comment

Validity of the findings

No comment

Additional comments

Thank you for the responses to my comments and clarifications/changes to the text. The manuscript has much improved.

Reviewer 3 ·

Basic reporting

The study can objectively answer your experimental question.

Experimental design

I believe that the description meets the parameters for the application of the study

Validity of the findings

The current study aimed to assess sex differences in pain using behavioral responses and an emotional reactivity test as a measure of emotional state. It has great regional implications and brings important discussions about the quality of management of sheep herds and animal welfare.

Additional comments

No more comments.